# Noninvasive Assessment of Fibrosis Following Ischemia/Reperfusion Injury in Rodents Utilizing Na Magnetic Resonance Imaging

**DOI:** 10.3390/pharmaceutics12080775

**Published:** 2020-08-14

**Authors:** Per Mose Nielsen, Christian Østergaard Mariager, Daniel Guldager Kring Rasmussen, Marie Mølmer, Federica Genovese, Morten Asser Karsdal, Christoffer Laustsen, Rikke Nørregaard

**Affiliations:** 1Department of Clinical Medicine, Aarhus University, 8200 Aarhus N, Denmark; permose@clin.au.dk (P.M.N.); mariemoelmer@gmail.com (M.M.); 2MR Research Centre, Department of Clinical Medicine, Aarhus University, 8200 Aarhus N, Denmark; cm@clin.au.dk (C.Ø.M.); cl@clin.au.dk (C.L.); 3Nordic Bioscience A/S, 2730 Herlev, Denmark; dgr@nordicbio.com (D.G.K.R.); fge@nordicbio.com (F.G.); mk@nordicbio.com (M.A.K.)

**Keywords:** renal IRI, MRI, rat models, interstitial fibrosis, noninvasive ECM markers

## Abstract

Fibrosis is often heterogeneously distributed, and classical biopsies do not reflect this. Noninvasive methods for renal fibrosis have been developed to follow chronic kidney diseases (CKD) and to monitor anti-fibrotic therapy. In this study, we combined two approaches to assess fibrosis regression following renal ischemia-reperfusion injury (IRI): magnetic resonance imaging (MRI) and noninvasive extracellular matrix (ECM) biomarkers. MRI was used to evaluate fibrosis in bilateral IRI in rats after reperfusion at 7, 14, and 21 days. This was performed with ^1^HT_1_ and T_2_* mapping, dynamic contrast-enhanced (DCE)-MRI, and chemical shift imaging (CSI)-^23^Na. The degradation of laminin gamma-1 chain (LG1M) and type III collagen (C3M) was measured in urine and plasma. Fibrosis was analyzed in tissue using fibronectin (FN) and alpha-smooth muscle actin (α-SMA) using quantitative polymerase chain reaction qPCR and western blotting. We found increased fibrosis 7 days after reperfusion, which dropped to sham levels after 21 days. Single kidney glomerular filtration rate (skGFR), perfusion (DCE-MRI), and total ^23^Na kidney content correlated positively with fibrotic markers FN and α-SMA as well as noninvasive LG1M and C3M. We showed that novel MRI protocols and ECM markers could track fibrogenic development. This could give rise to a multi-parametric practice to diagnose and assess fibrosis whilst treating kidney disease without using invasive methods.

## 1. Introduction

Chronic kidney diseases (CKD) lead to renal fibrosis and ultimately kidney failure. CKD is a leading cause of mortality, and its prevalence is steadily increasing globally, currently affecting about 10% of the population [1]. Renal fibrosis is a typical observed outcome in which the main characteristic is the increase of myofibroblasts activity in the renal interstitium producing extracellular matrix (ECM) deposition and accumulation [2].

Renal fibrosis assessment typically requires an invasive kidney biopsy causing several risks, e.g., bleeding. Since fibrosis is often heterogeneously distributed, a biopsy analysis is therefore inherently subjected to sampling bias [3]. Thus, there is a need for noninvasive methods to assess renal fibrosis in order to follow the disease development of CKD and to monitor anti-fibrotic therapy. In this study, we investigate two noninvasive approaches to evaluate renal fibrosis regression following ischemia/reperfusion injury, namely magnetic resonance imaging (MRI) and noninvasive ECM biomarkers.

Noninvasive methods such as diffusion-weighted (DW)-magnetic resonance imaging (MRI), blood oxygen level-dependent (BOLD)-MRI, T_1_ mapping, and renal MRI elastography have been proposed as novel methods to provide insight into renal function and fibrosis severity [4,5,6]. Accumulation of ECM components is believed to cause reduced water movement, and DW-MRI allows detection of Brownian motion of water; i.e., a higher DW-MRI signal may reflect the extent of fibrosis [7,8]. Complementarily, T_1_ mapping is a parametric plot where each pixel of a kidney image represents the ^1^H T_1_ spin-lattice of longitudinal relaxation time. ^1^H T_1_ relaxation time is mainly dependent on the amount of water molecules in the tissue. During renal fibrosis, a relaxation shift is likely to occur, which may be detected by T_1_ mapping [9]. Concomitantly, CKD patients have been shown to suffer from chronic hypoxia in the kidneys [10] and the inverse relationship with blood oxygenation assessed with the BOLD imaging may detect fibrotic lesions [11,12]. Magnetic elastography is an MR technique which in its essence enables the user to measure tissue “stiffness”. It is believed that kidney fibrosis elevates stiffness, leading to a higher signal [13].

Here, we aim to combine several MRI approaches, including not only the well-known sequences ^1^H T_1_ mapping and BOLD MRI but also dynamic contrast enhanced (DCE) MRI and ^23^Na chemical shift imaging (CSI) to assess renal function and the progression of renal fibrosis in an ischemia-reperfusion injury (IRI) disease model and to compare these to the results from a unilateral ureteral obstruction (UUO) model. A dynamic contrast enhanced (DCE)-MRI sequence allows to calculate perfusion and single-kidney glomerular filtration rate (skGFR) [14]. Currently, estimated GFR (eGFR) is the most important parameter used to diagnose CKD [15]. For this reason, the ability to assess skGFR is thought to give relevant information on kidney function and the related fibrotic state. Moreover, combining ^23^Na chemical shift imaging (CSI)-MRI might be an extra valuable tool to follow renal fibrosis [16]. A recent study has provided evidence that ^23^Na-MRI might be a promising tool to assess skin fibrosis by quantifying Na^+^ in the skin [16]. In this scenario, we utilize a similar approach to evaluate the signal intensity of ^23^Na in the whole kidney. In addition, we will evaluate ^23^Na signal intensity throughout segments in the kidney, allowing us to calculate the ^23^Na signal slope reflecting the osmotic gradient in the kidney. The use of ^23^Na mapping has not previously been evaluated for fibrosis quantification in the kidney, and this will, to our knowledge, be the first study to investigate this potential.

To complement our MRI data and to improve the understanding of pathophysiological alterations in tissue turnover following ischemia-reperfusion injury, we also measured novel noninvasive ECM markers in plasma and urine. Kidney fibrosis changes include type III collagen (COL III) accumulation in the tubulointerstitial matrix [17,18]. During disease progression, the altered microenvironment in the glomerular basement membrane (GBM) leads to an increased degradation of one of its most important components, namely laminin-11, which consists of the alpha-5, beta-2, and gamma-1 chain (LAMC1). The markers reflecting turnover of COL III (C3M) and LAMC1 (LG1M) measured in our study are both generated by matrix metalloproteinase-9 (MMP-9), a protease known to have a temporal and spatial expression during renal disease progression [19]. COL III degradation was assessed using the C3M enzyme-linked immunosorbent assay (ELISA), and LAMC1 degradation was assessed using the LG1M ELISA. The C3M biomarker has been previously evaluated in CKD animal models, where it correlated with the extent of renal fibrosis [17,20], while the LG1M, a LAMC1 biomarker, is still at early investigational stages [21].

Here, we compared different and innovative MRI techniques with C3M and LG1M measurements to improve the understanding of pathophysiological alterations in tissue turnover during kidney fibrosis development.

## 2. Materials and Methods

### 2.1. Animal Models

All experiments were performed on male Wistar rats divided into the following groups: sham *n* = 6; 240–290 g; 7 days IRI *n* = 6; 210–240 g; 14 days IRI *n* = 4; 235–260 g; 21 days *n* = 5; 300–330 g; and unilateral ureteral obstruction (UUO) *n* = 6; 200–230 g. These animals have been used in our previous paper published by Rasmussen et al. [22]. Some data was intentionally reused, and some are presented as originals, mainly the LG1M and MRI data. Two animals from two weeks IRI and one animal from three weeks IRI were euthanized prematurely. Animals had ad libitum access to standard rodent diet (Altromin, Lage, Germany) and water, and the 12 h:12 h light–dark cycle as well as temperature (21 ± 2 °C) and humidity (55 ± 5%) were controlled. The studies were carried out in accordance with the Danish National Guidelines for animal care and were approved by the Danish Animal Experiments Inspectorate under the Danish Veterinary and Food Administration (license no. 2015-15-0201-00658 for UUO animals and license no. 2013-15-2934-00810 for IRI animals).

The animals were anesthetized with sevoflurane (induction 6%, sustained 2.5%; Abbot Scandinavia, Solna, Sweden) mixed with air (2 L/min). Soon before the surgery began, Temgesic^®^ (Reckitt Berkshire, Slough, UK) was given subcutaneously as analgesics (0.05 mg/kg) and was then supplied in the drinking water (0.3 mg/mL) for 3 days post-surgery. During surgery, the animals were placed on a heating pad (CMA 450 temperature controller, Harvard apparatus, Cambourne, United Kingdom) to maintain a rectal temperature of 37 °C, while the respiration was visually monitored.

IRI induction: the abdomen was shaved and cleaned with ethanol, a surgical incision was made, and both the left and right renal arteries were carefully dissected. A nontraumatic clamp was placed on both arteries for 45 min to induce ischemia, where after the clamps were released, restoring blood flow and reperfusion visually were confirmed. To maintain postoperative water balance, 2 mL of isotonic saltwater was injected subcutaneously at the beginning of the IRI operation.

UUO induction: the abdomen was shaved and cleaned with ethanol, a surgical incision was made, and the left ureter was exposed and occluded with a 3-0 silk ligature. The incision was sutured separately in muscle tissue and skin. Sham animals underwent the same procedures, meaning dissection of either the renal artery or the left ureter but with no occlusion in both sham surgical cases.

After MR scanning, the IRI animals were euthanized. The UUO animals were scanned at both day 5 and day 7 and then euthanized. Arterial blood and urine were collected to estimate plasma creatinine and blood urea nitrogen (BUN). Creatinine and urea in plasma were measured using a Roche Cobas 6000 analyzer (Roche Diagnostics, Hvidovre, Denmark).

### 2.2. Fumarase Activity Assay

Fumarase activity was measured in plasma according to the manufacturer’s instructions (Sigma Aldrich, Brøndby, Denmark) with minor alterations, i.e., the analysis was performed in a 384-well costar plate using a PHERAstar FS microplate reader (BMG Labtech, Birkerød, Denmark). Absorbance readout was performed at the highest peak change (654 nm). Plasma was distributed without pretreatment.

### 2.3. Western Blotting (WB)

Renal cortex proteins were lysated in a dissecting buffer (0.3 M sucrose, 25 mM imidazole, and 1 mM (EDTA), pH 7.2) including protease inhibitors Complete Mini Protease Inhibitor Cocktail Tablets (Roche Diagnostics, Hvidovre, Denmark) and Phosphatase Inhibitor Cocktail 2 and 3 (Sigma-Aldrich, Brøndby, Denmark) using a tissue homogenizer (Qiagen, Hilden, Germany) followed by centrifugation. The total protein concentration was determined using a Pierce BCA protein assay kit (Roche Diagnostics, Hvidovre, Denmark) following the manufacturer’s instructions.

Proteins were separated on a 12% Criterion TGX Precast Gel (Bio-Rad Laboratories, Copenhagen, Denmark) and transferred to a Hybond ECL nitrocellulose membrane (GE Healthcare, Hatfield, UK). The membrane was then blocked in 5% non-fat dry milk in PBS-T (80 mM Na_2_HPO_4_, 20 mM NaH_2_PO_4_, 100 mM NaCl, and 0.1 Tween 20, pH 7.4), washed in PBS-T, and incubated with primary antibodies overnight at 4 °C. Subsequently, the membrane was again washed and incubated with HRP-conjugated secondary antibody for one hour at room temperature. Antigen-antibody reactions were visualized using a chemiluminescence system (Amersham ECL Plus, GE Healthcare). All western blots were normalized to total protein content measured with Stain-Free technology [23]. Primary and secondary antibodies are listed in Table 1.

### 2.4. Histology, Immunofluorescence (IF), and Immunohistochemistry (IHC)

A 2-mm kidney section was dissected from the IRI and sham kidneys when the rats were sacrificed. The sections were fixed in 4% paraformaldehyde for 2 hours and washed 3 times (10 min) with 0.01 M PBS buffer. The fixed kidney sections were then dehydrated, embedded in paraffin, and cut into 2-µM sections on a rotary microtome (Leica Microsystems, Herlev, Denmark). To assess the renal morphology, paraffin embedded kidney sections were stained with Hematoxylin and Eosin (HE). Renal fibrosis was highlighted by Masson’s trichrome, in which collagenous fibers were stained blue and the cytoplasm was stained red. The interstitial fibrosis degree was assessed by a renal pathologist examining the sections in a blinded manor and scoring the sections with either no fibrosis, mild fibrosis (1–5%), incipient fibrosis (6–10%), profound fibrosis (11–25%), severe fibrosis (26–50%), or advanced fibrosis (51–100%).

For IF, paraffin-embedded sections were stained with α-SMA (DAKO, Glostrup, Denmark, Cat. No. M0851) and DAPI (Thermofisher, Waltham, MA, USA, Cat. No. D1306) and was visualized with Alexa 488-conjugated secondary antibody (Thermofisher). Representative images are shown in 4× magnification. Fluorescence microscopy was accomplished using an Olympus BX61 microscope (Olympus, Tokyo, Japan). For IHC, the sections were stained with fibronectin (Abcam, Cat. No. ab2413), conjugated with secondary antibody P488 and goat anti-rabbit immunoglobin (DAKO), and counterstained with hematoxylin. Light microscopy was carried out using an Olympus BX50 microscope and CellSens imaging software (Olympus, Tokyo, Japan).

### 2.5. RNA Extraction and Quantitative PCR (qPCR)

Total RNA was extracted from renal cortex using NucleoSpin RNA II mini kit according to the manufacturer’s instructions (AH diagnostics, Aarhus, Denmark). Quantification of total RNA was performed by spectrophotometry and stored at −80 °C. cDNA synthesis was performed with RevertAid First strand cDNA synthesis kit following the manufacturer’s protocol (MBI Fermentas, Burlington, ON, Canada). qPCR was carried out using Maxima SYBR Green qPCR Master Mix according to the manufacturer’s instructions (AH diagnostics, Aarhus, Denmark) on an AriaMx QPCR reader (Agilent, Santa Clara, CA, USA). Briefly, 100 ng cDNA was utilized as template for PCR amplification. Specificity of products was confirmed by melting curve analysis and by gel electrophoresis. Primer sequences used are given in Table 2.

### 2.6. MR Scanning

Throughout MR examination, the animals were anesthetized with sevoflurane (2.5% sevoflurane and 2 L/min air). Rectal temperature, capillary oxygen saturation, and respiration were monitored throughout the MR session. A tail vein catheterization was performed for injection of the contrast agent. MRI examination was performed in a 9.4 T preclinical MR system (Agilent, Santa Clara, CA, USA) equipped with a dual tuned ^1^H/^23^Na rat volume coil (Doty Scientific, Columbia, SC, USA). The MRI examination included the following sequences: 1) A ^1^H gradient echo multi slice (GEMS) sequence was used for acquiring coronal and axial images (coronal: TR/TE = 60 ms/3.8 ms, flip angle = 20°, FOV = 60 × 120 mm^2^, matrix = 128 × 256; axial: TR/TE = 100 ms/2.6 ms, flip angle = 20°, FOV = 60 × 60 mm^2^, matrix = 128 × 128) and was used as an anatomical scout covering both kidneys; 2) for T_1_ measurements, a single-slice segmented look–locker sequence with a gradient-echo readout was used to acquire T_1_-weighted data (matrix = 128 × 128; FOV = 60 × 60 mm^2^; flip angle = 8°; TR/TE = 3.8 ms/1.9 ms; TI = 110, 180, 310, 510, 860, 1430, 2400, and 4000 ms; and 3-mm slice thickness); 3) ^1^H BOLD MRI was performed using a standard multi-echo gradient echo sequence with 10 echoes (TR/TE = 124 ms/2.4 ms, ΔTE = 2.2 ms, FOV = 60 × 60 mm^2^, matrix =128 × 128, flip angle = 30°, and 6-mm slice thickness; 4) T_2_*-weighted DCE was performed using an axial ^1^H gradient-echo sequence covering both kidneys in one slice (matrix = 128 × 128, FOV = 60 × 60 mm^2^, flip angle = 15°, TR/TE = 13.7 ms/1.9 ms, and 2-mm slice thickness). A single bolus of 50 µL of gadoteric acid (Dotarem^®^; 279.3 mg/mL; Guerbet, Villepinte, France) in isotonic saline (total injection volume of 1mL) was administered over 10 s; and 5) a 2D chemical shift imaging (CSI) sequence was performed to investigate ^23^Na (TR/TE = 60 ms/0.65 ms, flip angle = 90°, spectral width = 8 kHz, matrix = 32 × 32, FOV = 70 × 70 mm^2^, and 100 mm slice thickness).

### 2.7. MRI Analysis

MR images were converted to DICOM. Regions of interest (ROI) were drawn and analyzed in the commercial software OsiriX (Pixmeo SARL, Bernex, Switzerland). The mean ^1^H T_1_, ^1^H T_2_*, or ^23^Na T_2_* was calculated as the mean of all pixels included in the ROI. SkGFR calculated from DCE images was processed and analyzed in the commercial software Matlab (Mathworks, Natick, MA, USA). SkGFR was calculated using the Baumann–Rudin model. Perfusion changes from DCE wer calculated using the plugin UMMperfusion for Osirix, utilizing a fast deconvolution model. From the CSI images, a map of ^23^Na signal intensity was reconstructed. An ROI was drawn around the whole kidney and was normalized to a back-muscle ROI, giving rise to the total ^23^Na signal intensity kidney value. In order to reduce the user bias in the ROI placement, the novel twelve-layer concentric objects method (TLCO) [24] was adjusted to rodent MRI examinations, allowing correct renal assessment of the intrarenal modifications for all MRI examinations. In short, six equidistantly separated segment layers (12 layers in the original human version) were calculated based on a user-selected whole kidney ROI perimeter, which enabled us to calculate the ^23^Na slope through the segments, reflecting the osmotic kidney gradient.

### 2.8. Noninvasive ECM Markers

Levels of the noninvasive degradation markers of COL III (C3M) and LAMC1 (LG1M) in serum and urine from the animals were measured using competitive enzyme-linked immunosorbent assays (ELISA) developed by Nordic Bioscience, Denmark. Both of the monoclonal antibodies employed in the C3M and LG1M ELISA specifically detect neoepitopes of 10 amino acids in COL III (610′.KNGETGPQGP′619) and LAMC1 (1232′LNRKYEQAKN. ′1241), both generated by MMP-9. Intra- and inter-assay variations of the assays were <10% and <15%, respectively. The tests were carried out as follows. Briefly, a streptavidin-coated 96-well ELISA plate (cat. 11940279, Roche) was coated with a biotinylated peptide. The plate was washed in washing buffer five times and afterwards incubated with standard peptide or sample together with HRP-conjugated monoclonal antibody. Subsequently, the plate was again washed five times, followed by incubation of TMB (Kem-En-Tec, Taastrup, Denmark) in the dark. To end the reaction, a 1% sulfuric acid (H_2_SO_4_) solution was added and the plate was analyzed on the ELISA reader at 450 nm with 650 nm as the reference. All incubation steps were performed with constant shaking at 300 rpm. Concentrations were corrected for dilution factor of the samples. To normalize for urine output, biomarker levels in urine were divided by urinary creatinine levels.

### 2.9. Statistics

All data are presented as mean ± SEM. Normality was assessed with quantile plots. Differences were analyzed using a one-way analysis of variance (ANOVA) with a multiple comparison test comparing all IRI and UUO groups with sham and using a Bonferroni correction. Associations between variables were assessed by Pearson’s correlations analysis including both sham and IRI animals. Statistical analysis was performed in GraphPad PRISM 6 (GraphPad Software, San Diego, CA, USA) when comparing the different treatment groups. All *p*-values below 0.05 were considered statistically significant.

## 3. Results

### 3.1. Bilateral Ischemia Reperfusion Injury (IRI) and UUO Induced Kidney Injury

Rats were exposed to 45 min of bilateral ischemia followed by 7, 14, and 21 days of reperfusion. Similarly, a group of animals was exposed to UUO for 7 days. Bilateral renal IRI-induced plasma creatinine and blood urea nitrogen (BUN) increase peaked at 7 days after reperfusion and then normalized at 21 days (Figure 1A,B). The UUO rats indicated a tendency towards elevated values of plasma creatinine and BUN, although it did not reach significance. Fumarase activity, an indicator of necrosis development [25,26] was found to be elevated in the 7 days postischemic group and this elevation declined over time and normalized to sham levels in the 21 days postischemic group. No elevation of fumarase activity was found in the UUO group (Figure 1C). To evaluate kidney injury, we measured kidney injury marker 1 (KIM-1) and neutrophil gelatinase-associated lipocalin (NGAL). KIM-1 mRNA expression was increased after 7 days bilateral IRI and in rats subjected to UUO, whereas no significant change was observed at day 14 and day 21 after IRI (Figure 1D). NGAL mRNA expression was significantly increased in the UUO rats and elevated in the 7 days and 14 days postischemic kidney, but it did not reach statistical significance (Figure 1E).

### 3.2. Bilateral IRI and UUO Induced Renal Fibrosis

In order to evaluate progression of fibrosis after IRI and UUO, we measured the fibrosis markers alpha-smooth muscle actin (α-SMA) and fibronectin (FN) in kidney cortex. α-SMA and FN mRNA expression as well as protein levels were significantly increased 7 days after reperfusion in rats subjected to bilateral IRI and UUO (Figure 2A–D). In addition, we also observed a significant increase in FN mRNA expression after 14 days IRI (Figure 2B). No change in both mRNA and protein levels of α-SMA and FN were observed after 21 days of reperfusion. Representative immunofluorescence staining of α-SMA and FN immunoperoxidase staining confirmed our mRNA and protein data, showing stronger staining intensity at 7 and 14 days after IRI (Figure 2F,G,J,K respectively) compared to the sham group (Figure 2E,I) and the IRI groups with a longer reperfusion time (Figure 2G,H,K,L).

### 3.3. Changes in the Level of ECM Markers C3M and LG1M in Response to Bilateral IRI

The biomarker levels of C3M and LG1M were investigated in the urine and plasma fraction of the bilateral IRI groups. Levels of urinary C3M (U-C3M) closely mirrored the fibrotic response with increased presence of the biomarker at 7 and 14 days after reperfusion and returning to baseline levels at 21 days after reperfusion (Figure 3A). We found no significant change in the plasma fraction of C3M (P-C3M) (Figure 3B). LG1M in urine (U-LG1M) was increased at 14 days of reperfusion, whereas no significant change was observed in the plasma LG1M (P-LG1M) (Figure 3C,D) although a trend towards increased plasma levels was observed at 14 days after reperfusion.

### 3.4. MRI Images, Values, and Correlations with Fibrosis Markers and Noninvasive ECM Markers

The MRI protocols were adapted for a 9.4 T rodent scanner. The skGFR and perfusion calculated from the DCE images were significantly lower at 7 days of reperfusion and in UUO rats scanned at both day 5 and day 7 (Figure 4A,B). The total ^23^Na signal intensity/kidney was found to be significantly elevated at 7 days of reperfusion (Figure 4C). In addition, the kidney ^23^Na slope was significantly reduced at 7 days of UUO (Figure 4D). Representative ^23^Na-MRI images of the kidneys were obtained in all the animals (Figure 4E–J). MRI with 1H T1 and T2* mapping did not show any significant change among the groups (data not shown). 

To investigate the correlation between fibrosis and our MRI data, injury markers, and noninvasive ECM markers, we performed a Pearson’s correlation analysis of all groups. Assessment of fibrosis was performed using HE (Figure 5A–D) and Masson’s trichrome staining (Figure 5E–H) as previously published^21^ and now used for further analyses related to our MRI data. The fibrotic score was significantly elevated after 7 and 14 days of reperfusion (Figure 5I), and no change was seen after 21 days. We found a negative correlation between fibrosis score and skGFR (Figure 5J) and a positive correlation between fibrosis score and total ^23^Na signal (Figure 5K). No significant correlation was found in relation to ^23^Na signal slope (Figure 5L).

MRI measurements of skGFR, perfusion, and total ^23^Na have a positive correlation with mRNA of α-SMA and FN (Figure 6A–C). The ^23^Na signal slope has a negative correlation with mRNA of α-SMA and FN (Figure 6A,D). In addition, U-C3M, P-C3M, and P-LG1M were associated with increased mRNA levels of α-SMA (Figure 6E–G). U-LG1M had no significant association with α-SMA and FN mRNA levels.

After establishing possible fibrosis correlations, we further investigated correlations between the MR data (skGFR, perfusion, total ^23^Na signal slope, and total ^23^Na signal) and our noninvasive ECM markers (Figure 7A). MRI measures of skGFR and total ^23^Na signal slope as well as total ^23^Na signal correlated inversely with U-C3M (Figure 7B,D,E). No correlation was found for perfusion and U-C3M (Figure 7C). P-C3M correlated with skGFR and perfusion as well as total ^23^Na signal (Figure 7F–I). No correlation was found for ^23^Na slope (Figure 7H). In addition, a correlation between ^23^Na slope and P-LG1M was observed (Figure 7A).

## 4. Discussion

The main finding of this study is that skGFR, perfusion, ^23^Na slope, and total ^23^Na signal intensity as estimated by DCE imaging and CSI were correlated with renal function and fibrosis regression in both IRI and UUO models while MR relaxometry did not show any association. In addition, the noninvasive ECM urinary biomarker C3M correlated with the fibrogenic response as well as MRI. Measured skGFR and total ^23^Na in the renal cortex were able to noninvasively reflect the fibrotic response following IRI in rats.

We used two different animal models, unilateral ureteral obstruction (UUO) and bilateral ischemia reperfusion injury (IRI), to follow fibrosis progression and recovery. The UUO model is a well-known fibrosis model in rodents [27]. The bilateral IRI model is a more complex one where strict control of the severity of the induced ischemic renal injury is critical. If the induced ischemic insult is too severe, animals are likely to die within 48 h [28].

We used 45 min of ischemia to ensure that the kidneys could recover again; thus, we were able to test our noninvasive approach in the fibrogenic response and during recovery of the kidney. Interestingly, fibrosis markers and fibrosis scoring were highly elevated in the early reperfusion phase, followed by a drop towards sham levels after 21 days. The noninvasive extracellular matrix (ECM) biomarker reflecting urinary degradation fragments of COL III (U-C3M) closely followed the fibrosis scoring with the highest elevation after 7 days of reperfusion, remaining elevated at 14 days and dropping to sham levels after 21 days of reperfusion, supporting the restructuring of the ECM. Laminin gamma chain-1 (LG1M) levels in both plasma and urine peaked after 14 days of reperfusion and then dropped to near sham levels at 21 days of reperfusion. ECM turnover during fibrogenesis is a dynamic process, with different rates of formation and degradation of ECM proteins [29]. Our findings for both U-C3M and LG1M support this temporal turnover. Together, this indicates that the bilateral IRI model seems to be a quite useful injury model as it is reversible compared to the UUO. It is important to note that blood urea nitrogen (BUN), creatinine, and fumarase were not significantly elevated in UUO, which is likely caused by the reduced perfusion. Renal injury markers (KIM-1, NGAL, and fumarase) were similarly found to correlate with single-kidney filtration glomerular rate (skGFR), perfusion, and ^23^Na-chemical shift imaging (CSI). It was not possible, though, to separate the effects on renal function imposed by fibrosis alone using MRI methods.

This study confirms previous findings that renal function is directly coupled to the level of renal damage and fibrosis [30]. This was particularly evident from the evaluation of renal function using two independent sequences, including dynamic contrast enhanced (DCE; skGFR) and CSI (total ^23^Na signal and signal slope). This gives rise to a more reliable set of markers of kidney dysfunction than MR relaxometry, which is inherently based on the magnetic properties and molecular environment of water in the investigated tissue. This means that all factors affecting water balance will be detected in the MR sequence, including perfusion changes, blood flow, edema, kidney growth, blood volume, GFR, diuresis, and ECM composition [5]. Naturally, we cannot rule out the possibility that the quality, sensitivity, and resolution of our images is too poor to utilize relaxometry. However, the resolution of our DCE and CSI images show significant changes with good robustness in spite of a potentially reduced image quality. Interestingly, total ^23^Na kidney content also reached a significant change and showed strong correlation with fibrosis regression in the kidney as well as correlation with the COL III degradation marker in the urine (U-C3M). Total ^23^Na kidney content is potentially independent of kidney function, giving rise to an independent measurement of fibrosis compared to skGFR and ^23^Na kidney slope.

Kidney fibrosis is often heterogeneously distributed, and in principle, MRI can differentiate the heterogeneous fibrosis; however, the current resolution was limited, and as such, the data presented are limited to rather large voxels. This is solely a limiting case in small animals, and as such, focal differentiation is possible in humans. All MRI methods can be applied in humans [31,32,33,34,35,36,37]. However, the use of gadolinium contrast agents is contradicted in severe CKD, and as such, alternative methods would likely be used to determine the hemodynamic nature of kidneys in humans [38]. 

The ability to noninvasively visualize kidney function and specifically pinpoint areas giving rise to the reduced kidney function makes the ^23^Na content and/or slope an intriguing new MRI method that may be used to monitor renal fibrosis. These MRI measurements can either alone or in combination with noninvasive ECM markers, such as C3M, be a valuable tool to assess fibrosis regression in CKD patients where biopsy may not be recommended. This means that ECM markers and MR data potentially can give rise to a completely noninvasive procedure to evaluate fibrosis in a variety of renal diseases. Such noninvasive procedures may also be used in drug development to better assess the effect of reno-protective or anti-fibrotic drugs on structural aspects of the kidney. This would greatly improve clinical trials that currently mostly focus on change in renal function (serum creatinine) and/or albuminuria. Therefore, this study highlights the importance of incorporating multi-parametric information in the stratification of renal injury and fibrosis.

## 5. Patents

The patents for the C3M and LG1M ELISAs are owned by Nordic Bioscience.

## Figures and Tables

**Figure 1 pharmaceutics-12-00775-f001:**
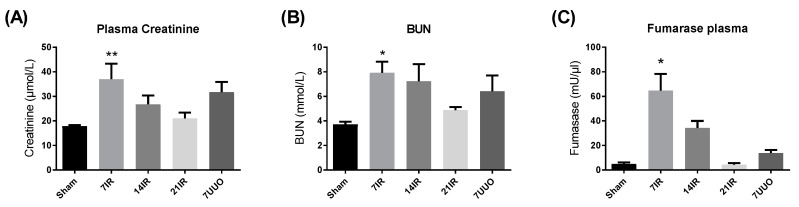
**Kidney injury induction after ischemia reperfusion (IR) injury and unilateral ureteral obstruction (UUO):** (**A**) plasma creatinine, (**B**) blood urea nitrogen (BUN), (**C**) fumarase activity, (**D**) kidney injury molecule-1 (KIM-1) mRNA expression corrected for 18S ribosomal protein, and (**E**) neutrophil gelatinase-associated lipocalin (NGAL) mRNA expression corrected for 18S ribosomal protein. ANOVA with Bonferroni multiple comparison test was performed between sham, ischemia-reperfusion injury (IRI) and UUO. Each bar represents the mean ± SEM, * *p* < 0.05 compared to sham group (*n* = 6). ** *p* < 0.01 compared to sham group. **7IR**: 7 days of reperfusion (*n* = 6), **14IR**: 14 days of reperfusion (*n* = 4), **21IR**: 21 days of reperfusion (*n* = 5), and **7UUO**: 7days UUO model (*n* = 6).

**Figure 2 pharmaceutics-12-00775-f002:**
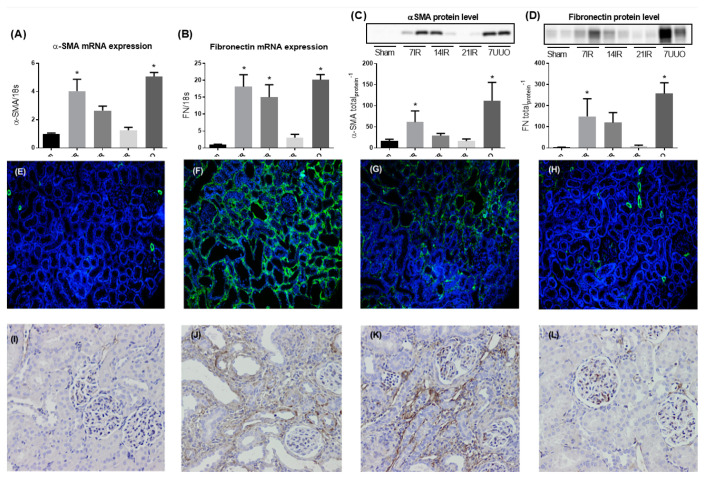
**Fibrosis markers in renal tissue:** (**A**) mRNA expression of αSMA and (**B**) fibronectin both normalized to the ribosomal protein 18S; (**C**) western blot analysis of αSMA, including band 42 kDa, normalized to total protein content; and (**D**) western blot of fibronectin, including band 260 kDa, normalized to total protein content. ANOVA with Bonferroni multiple comparison test was performed between sham, IRI, and UUO. Each bar represents the mean ± SEM, * *p* < 0.05 compared to sham group. Representative αSMA immunofluorescence (IF) stain of (**E**) sham (*n* = 6), (**F**) 7 days after bilateral IRI (*n* = 6), (**G**) 14 days after bilateral IRI (*n* = 4), and (**H**) 21 days after bilateral IRI (*n* = 5): Similarly, representative FN immunohistochemistry (IHC) stains of (**I**) sham, (**J**) 7 days after bilateral IRI, (**K**) 14 days after bilateral IRI, and (**L**) 21 days after bilateral IRI. IF magnification 4×, IHC magnification 10×.

**Figure 3 pharmaceutics-12-00775-f003:**
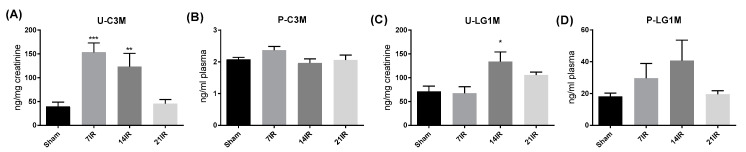
**Noninvasive extracellular matrix (ECM) markers; type III collagen (C3M); and alpha-5, beta-2, and gamma-1 chain (LAMC1) in urine and plasma:** (**A**) urine C3M (U-C3M), (**B**) plasma C3M (P-C3M), (**C**) urine LG1M (U-LG1M), and (**D**) plasma LG1M (P-LG1M). ANOVA with Bonferroni multiple comparison test was performed between sham and IRI. Each bar represents the mean ± SEM, * *p* < 0.05 compared to sham group (*n* = 6). ** *p* < 0.01 compared to sham group. *** *p* < 0.001 compared to sham group. 7IR: 7 days of reperfusion (*n* = 6), 14IR: 14 days of reperfusion (*n* = 4), and 21IR: 21 days of reperfusion (*n* = 5).

**Figure 4 pharmaceutics-12-00775-f004:**
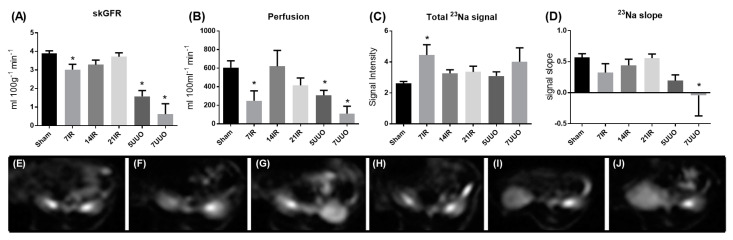
**MRI data:** (**A**) single-kidney glomerular filtration rate (skGFR) values calculated from the dynamic contrast enhanced (DCE) sequence, (**B**) perfusion calculated from the DCE sequence, (**C**) total ^23^Na signal in the kidney normalized to back muscle ^23^Na signals, and (**D**) slope of the ^23^Na signal from the cortex to inner medulla. ANOVA with Bonferroni multiple comparison test was performed between sham, IRI, and UUO. Each bar represents the mean ± SEM, * *p* < 0.05 compared to sham group (*n* = 6). Representative images of the ^23^Na signal from (**E**) the sham group (*n* = 6), (**F**) 7 days after bilateral ischemia reperfusion injury (IRI) (*n* = 6), (**G**) 14 days after bilateral IRI (*n* = 4), (**H**) 21 days after bilateral IRI (*n* = 5), (**I**) 5 days of unilateral ureteral obstruction (UUO) (*n* = 6), and (**J**) 7 days of UUO (*n* = 6).

**Figure 5 pharmaceutics-12-00775-f005:**
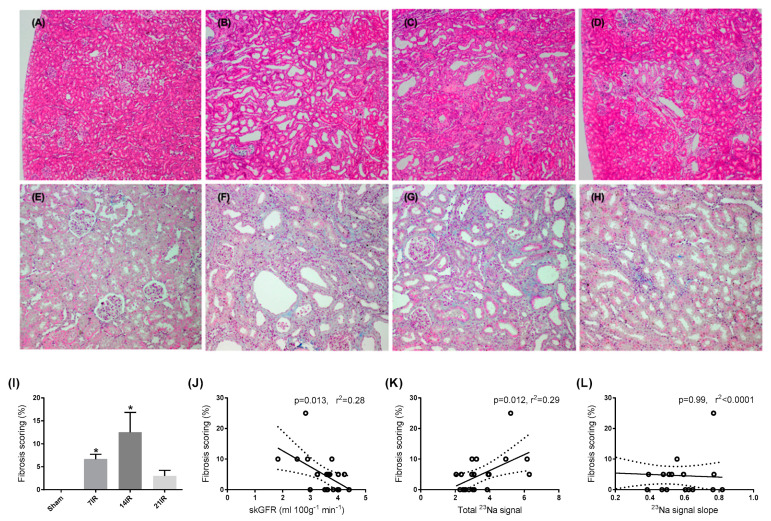
**Fibrosis scoring of renal tissue after bilateral ischemia reperfusion injury (IR) and its correlation with MRI data:** Representative hemoxylin and eosin (HE) stains of (**A**) sham (*n* = 6), (**B**) 7 days after bilateral IRI (*n* = 6), (**C**) 14 days after bilateral IRI (*n* = 4), and (**D**) 21 days after bilateral IRI (*n* = 5). Magnification 4×. Representative Masson’s trichrome of (**E**) sham, (**F**) 7 days after bilateral IRI, (**G**) 14 days after bilateral IRI, and (**H**) 21 days after bilateral IRI. Magnification 4×. (**I**) Fibrosis scoring based on HE and Masson’s trichrome staining. ANOVA with Bonferroni multiple comparison test was performed between Sham and IRI groups. Each bar represents the mean ± SEM. * *p* < 0.05 compared to sham group. (**J**) Correlation between fibrosis scoring and single-kidney glomerular filtration rate (skGFR), (**K**) correlation between fibrosis scoring and total ^23^Na signal, and (**L**) correlation between fibrosis scoring and ^23^Na signal slope.

**Figure 6 pharmaceutics-12-00775-f006:**
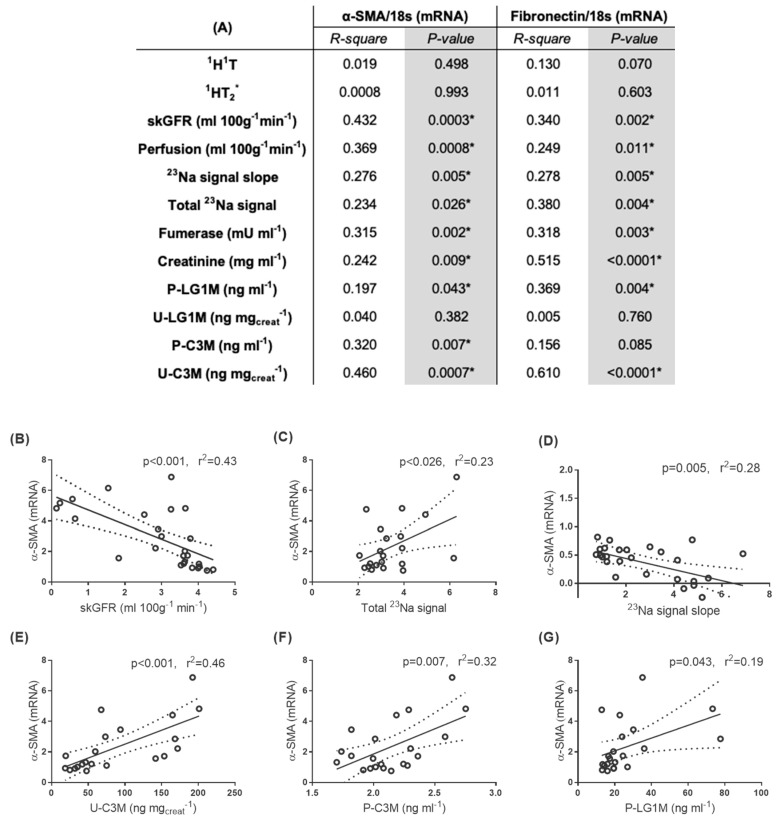
**Correlations of αSMA and fibronectin with MRI data and noninvasive extracellular markers as well as markers of kidney damage:** (**A**) a table with Person’s correlations values between tissue fibrosis markers (α-SMA and fibronectin) and utilized MRI sequences, kidney damage markers, and the noninvasive extracellular markers and linear correlations graphs (**B**) between α-SMA mRNA and single-kidney glomerular filtration rate (skGFR), (**C**) between α-SMA mRNA and total ^23^Na signal, (**D**) between α-SMA mRNA and ^23^Na signal slope, (**E**) between α-SMA mRNA and urinary type III collagen marker (U-C3M), (**F**) between α-SMA mRNA and plasma C3M (P-C3M), and (**G**) between α-SMA mRNA and plasma LG1M (P-LG1M). * *p* < 0.05 was considered significant.

**Figure 7 pharmaceutics-12-00775-f007:**
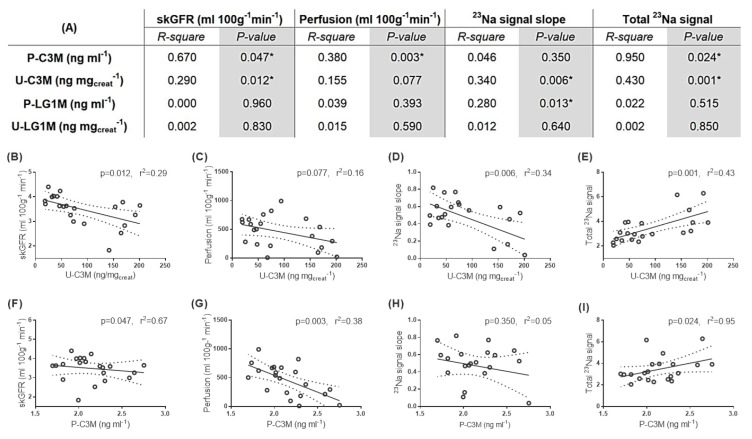
**Correlations of extracellular matrix (ECM) markers, type III collagen (C3M), and laminin gamma-1 chain (LG1M) with MRI data:** (**A**) a table with Person’s correlations values between noninvasive ECM fibrosis biomarkers (C3M and LG1M) in urine (U) and plasmatic (P) samples and the utilized MRI sequences: single-kidney glomerular filtration rate (skGFR), perfusion, total ^23^Na, and ^23^Na slope. Linear correlations graphs between (**B**) skGFR and U-C3M, (**C**) perfusion and U-C3M, (**D**) ^23^Na signal slope and U-C3M, (**E**) total ^23^Na signal and U-C3M, (**F**) skGFR and P-C3M, (**G**) perfusion and P-C3M, (**H**) ^23^Na signal slope and U-C3M, and (**I**) total ^23^Na signal and P-C3M. * *p* < 0.05 was considered significant.

**Table 1 pharmaceutics-12-00775-t001:** Primary and secondary antibodies for western blotting.

**Primary Antibodies**			
**Protein**	**Dilution**	**Company**	**Catalogue No.**
α-SMA	1:1000	DAKO, Glostrup, Denmark	M0851
Fibronectin (FN)	1:1000	Abcam, Cambridge, United Kingdom	ab2113
**Secondary Antibodies**			
**Antigen**		**Company**	**Catalogue No.**
Goat anti-mouse immunoglobulin/HRP		DAKO, Glostrup, Denmark	P447
Goat anti-rabbit immunoglobulin/HRP		DAKO, Glostrup, Denmark	P448

**Table 2 pharmaceutics-12-00775-t002:** Quantitative PCR (qPCR) primers.

Target Gene	Forward Primer Sequence	Reverse Primer Sequence
18s	5′ -CAT GGC CGT TCT TAG TTG-3′	5′-CAT GCC AGA GTC TCG TTC-3′
α-SMA	5′-CAT CAT GCG TCT GGA CTT GG-3′	5′-CCA GGG AAG AAG AGG AAG CA-3′
Fibronectin	5′-CCG AAT CAC AGT AGT TGC GG-3′	5′-GCA TAG TGT CCG GAC CGA TA-3′
KIM-1	5′-CCA CAA GGC CCA CAA CTA TT-3′	5′-TGT CAC AGT GCC ATT CCA GT-3′
NGAL	5′-GAT CAG AAC ATT CGT TCC AA-3′	5′-TTG CAC ATC GTA GCT CTG TA-3′

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
