# Peer review of "Noninvasive Assessment of Fibrosis Following Ischemia/Reperfusion Injury in Rodents Utilizing Na Magnetic Resonance Imaging"

_pharmaceutics, 2020, doi:10.3390/pharmaceutics12080775_

Round 1

Reviewer 1 Report

General Comment: This is an interesting study of the role of MRI to assess kidney fibrosis.

Major Comment: The authors frequently refer to fibrosis progression.  However their models do not assess progression.  UUO is measured at a single time point.  IRI is measured at 3 timepoints but fibrosis is maximal at 1 week and decreases thereafter.  Thus, there is no progression.  It would be more accurate to say that the paper assesses fibrosis or possibly fibrosis regression, but not progression.

Minor Comments:

1.  The authors comment that fibrosis is often heterogeneous.  Are they able to detect this heterogeneity by MRI?

2.  Which of the MRI techniques that they study in this paper can be used on humans?

Author Response

General Comment: This is an interesting study of the role of MRI to assess kidney fibrosis.

Major Comment: The authors frequently refer to fibrosis progression. However their models do not assess progression.  UUO is measured at a single time point. IRI is measured at 3 timepoints but fibrosis is maximal at 1 week and decreases thereafter. Thus, there is no progression. It would be more accurate to say that the paper assesses fibrosis or possibly fibrosis regression, but not progression.

ANSWER: We thank the reviewer for this important comment. We agree with the reviewer that we are not able to assess progression of fibrosis in the IRI model as it is maximal at one week. However, in the UUO model we have performed two MRI scans in the same rats in order to follow fibrosis progression. One scan at 5 days UUO and one more at 7 days UUO and our data show that fibrosis is more severe after 7dUUO compared to 5dUUO. We have now made this more clear in both the method and result sections. In addition, we have also changed fibrosis progression to fibrosis regression.

Minor Comments:

  1. The authors comment that fibrosis is often heterogeneous.  Are they able to detect this heterogeneity by MRI? ANSWER: In principle MR can differentiate the heterogenous fibrosis, however the current resolution was limited and as such the data presented are limited to rather large voxels. This is however a limit that is solely a limiting case in small animals and as such focal differentiation is possible in humans.
  2. Which of the MRI techniques that they study in this paper can be used on humans? ANSWER: All MRI methods can be applied in humans (1-7). However, the use of gadolinium contrast agents are contradicted in severe CKD and as such alternative methods would likely be used to determine the hemodynamic nature of the kidneys in humans (8).

References:

  1. Dekkers, I.A.; Boer, A.D.; Sharma, K.; Cox, E.F.; Lamb, H.J.; Buckley, D.L.; Bane, O.; Morris, D.M.; Prasad, P.V.; Semple, S.I.K.; Gillis, K.A.; Hocking, P.; Buchanan, C.; Wolf, M.; Laustsen, C.; Leiner, T.; Haddock, B.; Hoogduin, J.M.; Pullens, P.; Sourbron, S.; Francis, S. “Consensus-based technical recommendations for clinical translation of renal T1 and T2 mapping MRI.” MAGMA. 2020, 33, 163-76; DOI: 10.1007/s10334-019-00797-5.
  2. Ljimani, A.; Caroli, A.; Laustsen, C.; Francis, S.; Mendichovszky, I.A.; Bane, O.; Nery, F.; Sharma, A.; Pohlmann, A.; Dekkers, I.A.; Vallee, JP.; Derlin, K.; Notohamiprodjo, M.; Lim, R.P.; Palmucci, P.; Serai, S.D.; Periquito, J.; Wang, Z.J.; Froeling, M.; Thoeny, H.C.; Prasad, P.; Schneider, M.; Nienhorf, T.; Pullens, P.; Sourbron, S.; Sigmund E.E. “Consensus-based technical recommendations for clinical translation of renal diffusion-weigthed MRI.” MAGMA. 2020, 33, 177-95; DOI: 10.1007/s10334-019-00790-y.
  3. Nery, F.; Buchanan, C.E.; Harteveld, A.A.; Odudu, A.; Bane, O.; Cox, E.F.; Derlin, K.; Gach, H.M.; Golay, X.; Gutberlet, M.; Laustsen, C.; Ljimani, A.; Madhuranthakam, A.J.; Pedrosa, I.; Prasad, P.V.; Robson, P.M.; Sharma, K.; Sourbron, S.; Taso, M.; Thomas, D.L.; Wang, D.J.J.; Zhang, J.L.; Alsop, D.C.; Fain, S.B.; Francis, S.T.; Fernádez-Seara, M.A. “Consensus-based technical recommendations for clinical translation of renal ASL MRI.” MAGMA. 2020, 33, 141-61; DOI: 10.1007/s10334-019-00800-z.
  4. Mendichovszky, I.A.; Pullens, P.; Dekkers, I.A.; Nery, F.; Bane, O.; Pohlmann, A.; Boer, A.D.; Ljimani, A.; Odudu, A.; Buchanan, C.; Sharma, K.; Laustsen, C.; Harteveld, A.; Golay, X.; Pedrosa, I.; Alsop, D.C.; Fain, S.; Caroli, A.; Prasad, P.; Francis, S.; Sigmund E.E.; Fernádez-Seara, M.A.; Sourbron, S. ”Technical recommendations for clinical translation of renal MRI: a consensus project of the Cooperation in Science and Technology Action PARENCHIMA.” MAGMA. 2020, 33, 131-40; DOI: 10.1007/s10334-019-00784-w.
  5. Bane, O.; Mendichovszky, I.A.; Milani, B.; Dekkers, I.A.; Deux, J-F.; Eckerbom, P.; Greiner, N.; Hall, M.E.; Inoue, T., Laustsen C.; Lerman, L.O.; Liu, C.; Morrell, G.; Pedersen, M.; Pruijm, M.; Sadowski, E.A.; Seeliger, E.; Sharma, K.; Thoney, H.; Vermathen, P.; Wang, Z.J.; Serafin, Z.; Zhang, J.L.; Francis, S.; Sourbron, S.; Pohlmann, A.; Fain, S.; Prasad, P.V. “Consensus-based technical recommendations for clinical translation of renal BOLD MRI.” MAGMA. 2020, 33, 199-15; DOI: 10.1007/s10334-019-00802-x.
  6. Hockings, P.; Laustsen, C.; Joles, J.A.; Mark, P.B.; Sourbron, S. “Special issue on magnetic resonance imaging biomarkers of renal diseases.” MAGMA. 2020, 33, 1-2; DOI: 10.1007/s10334-019-00822-7. 
  7. Grist, J.T.; Riemer, F.; Hansen, E.; Tougaard, R.S.; McLean, M.A.; Kaggie, J.; Bøgh, N.; Graves, M.J.; Gallagher, F.A.; Laustsen, C. “Visualization of sodium dynamics in the kidney by magnetic resonance imaging by a multisite study.” Kidney International. 2020. DOI: https://doi.org/10.1016/j.kint.2020.04.056.
  8. Pedersen, M.; Ursprung, S.; Jensen, J.D.; Jespersen, B.; Gallagher, F.; Laustsen, C. “Hyperpolarised 13C-MRI metabolic and functional imaging: an emerging renal MR diagnostic modality.” MAGMA. 2020, 33, 23-33; DOI: 10.1007/s10334-019-00801-y.  

Reviewer 2 Report

The manuscript is well written, well documented and easy to follow. The topic was correctly put into perspective with existing studies. The authors provided a good background.
The interest to readers of "Pharmaceutics" is potentially high.

The authors compared innovative MRI techniques with C3M and LG1M measurements to improve the understanding of pathophysiological alterations in tisue turnover during kidney fibrosis development. The proposed label-free method represents a very good appoach. I suggest them only minor revisions, in particular in Materials and Methods section I would add the scan parameters used to perform the MRI acquisitions.

Author Response

The manuscript is well written, well documented and easy to follow. The topic was correctly put into perspective with existing studies. The authors provided a good background. The interest to readers of "Pharmaceutics" is potentially high.

The authors compared innovative MRI techniques with C3M and LG1M measurements to improve the understanding of pathophysiological alterations in tissue turnover during kidney fibrosis development. The proposed label-free method represents a very good approach. I suggest them only minor revisions, in particular in Materials and Methods section I would add the scan parameters used to perform the MRI acquisitions. 

ANSWER: We thank the reviewer for these positive comments. We have now included more details in the Materials and Methods section and all the scan parameters are described in details in the section 2.6.

The MRI examination included the following sequences: 1) A 1H gradient echo multi slice (GEMS) sequence was used for acquiring coronal and axial images (Coronal: TR/TE = 60ms/3.8ms, flip angle = 20°, FOV = 60×120mm2, matrix = 128×256. Axial: TR/TE = 100ms/2.6ms, flip angle = 20°, FOV = 60×60mm2, matrix = 128×128) was used as an anatomical scout covering both kidneys. 2) For T1-measurements, a single-slice segmented Look–Locker sequence with a gradient-echo readout was used to acquire T1-weighted data (matrix = 128×128, FOV = 60×60mm2, flip angle = 8°, TR/TE = 3.8ms/1.9ms, TI = 110, 180, 310, 510, 860, 1430, 2400, 4000 ms, and 3mm slice thickness). 3) 1H BOLD MRI was performed using a standard multi-echo gradient echo sequence with 10 echoes (TR/TE= 124ms /2.4ms, ΔTE = 2.2ms, FOV =60×60mm2, matrix=128×128, flip angle=30° and 6mm slice thickness. 4) T2*-weighted DCE was performed using an axial 1H gradient-echo sequence, covering both kidneys in one slice (matrix = 128×128, FOV = 60×60mm2, flip angle = 15°, TR/TE = 13.7ms/1.9ms and 2mm slice thickness). A single bolus of 50µL of Gadoteric acid (Dotarem®; 279.3mg/mL; Guerbet, Villepinte, FR) in isotonic saline (total injection volume of 1mL) was administered over 10s. 5) A 2D chemical shift imaging (CSI) sequence was performed to investigate 23Na (TR/TE = 60ms/0.65ms, flip angle = 90°, spectral width = 8kHz, matrix = 32×32, FOV = 70×70mm2 and 100mm slice thickness).